# Perception of urban green space among university students in Bangladesh

**Bijoya Saha, Shah Md Atiqul Haq** *

Department of Sociology, Shahjalal University of Science and Technology, Sylhet, Bangladesh

* shahatiq1@yahoo.com, shahatiq-soc@sust.edu

## Abstract

Public parks and other green areas are crucial components of urban development. Urban management in emerging countries such as Bangladesh faces major challenges, especially because of the socio-environmental impacts of urbanization. Urban management initiatives in developing countries sometimes neglect crucial services for university students, such as study environments and recreational facilities. This study aimed to investigate students' perceptions of urban green space (UGS) and its potential benefits to our daily lives at Shahjalal University of Science and Technology (SUST) in Sylhet, Bangladesh, We collected data from 438 respondents by using a survey questionnaire-based stated preference approach as a methodological tool, using non-monetary assertions as the basis for the method. The survey included questions about respondents' social background, the frequency with which they visited green spaces, the benefits of urban green spaces and their perception of urban green spaces. We performed statistical analysis both descriptive and inferential statistics. Our findings suggest that 71.7% of students primarily use street trees and peace gardens as their main sites for urban green spaces. The study suggests that the advantages of urban green spaces, which include physical, mental, and environmental benefits, are strongly associated with criteria such as gender, academic level, and vulnerable to home locality for climate change (CC) or extreme weather events (EWEs). The Binary Logistic regression analysis identified urban life as the most influential factor. The correct classification rate was approximately 74.7%, indicating the model's strong accuracy in classification. Students who have lived in urban areas for more than 20 years have a reduced awareness of urban green spaces at 5% level of significance. Planning and policymaking for the creation and administration of urban green spaces, considering aspects like land use and environmental sustainability, could benefit from this study.

**Data Availability Statement:** All relevant data are within the article and its Supporting Information files.

**Funding:** The author(s) received no specific funding for this work.

## 1 Introduction

These days, a thorough understanding of urban green spaces is essential. Researchers are increasingly interested in the health and well-being of people living in city centers, and they believe that urban green spaces could play a crucial role in promoting this well-being. However, this is not a new concept. The development of urban public parks in the 19th century

**Competing interests:** The authors have declared that no competing interests exist.

arose from a desire to improve the health and well-being of workers in newly industrialized cities and was based on the strong assumption that green open spaces would have special health benefits for the urban poor [1]. As well as being beneficial for a healthy lifestyle, urban green spaces can help combat climate change [2]. Cities are increasingly developing land for urban purposes, while trees can support surprisingly diverse ecosystems. Residents of post-industrial cities enjoy several benefits when there are more trees [3,4]. Urban green and blue spaces can improve the economy and biodiversity [5]. City ecosystems benefit from urban forests [3].

In Bangladesh, green spaces play an important role as social and cultural centers, providing venues for leisure activities, community meetings and cultural events. Nevertheless, many cities in Bangladesh lack green spaces per capita. According to World Bank figures, by 2023, over 40.47% of Bangladesh's population will reside in cities, making the function of urban forests even more crucial [6]. Rapid urbanization in developing countries like Bangladesh is damaging social, environmental and economic well-being [7] and reducing greenery [8]. Since many homes have been built on private land without permission, urban areas lack infrastructure to protect citizens from natural disasters and other damage [9]. Bangladesh's urban management strategies are strongly influenced by the effects of climate change, particularly with regard to natural disasters such as floods, cyclones and sea-level rise [8,10]. But residents' demands for administrative space and greenery are frequently disregarded. Every city in the world has urban green spaces. The World Bank suggests that the most effective mitigation option is to increase and enhance urban green areas [6]. Green areas in cities help with problem-solving. Greening has been seen as a critical component of urban populations' quality of life, particularly in light of scenarios of human-induced climate change and their consequences on temperature and health [11]. "Green infrastructure" refers to urban green space. It is an essential component of a city's public open spaces and common services, and it can enhance the well-being of all urban people. Thus, public green spaces must be assessed to ensure that they are accessible to everybody and evenly dispersed across the urban environment [12]. City greening includes squares, parks, and urban woods. The latter method has the potential to mitigate climate change while also protecting the ecosystem [4].

The notion of urban green spaces is widely known, but the perception of urban green spaces is a relatively new development for our age, particularly in Bangladesh. Bangladesh's growing population and urbanization frequently result in a decline in green spaces. Furthermore, urban planning in Bangladesh favors infrastructural development above green spaces [2]. The perception of urban green spaces is an important disciplinary backgrounds, especially at educational institutions like universities, where it is examined at several levels to address sustainability problems. This investigation frequently involves an analysis of how different educational disciplines see the significance and role of green areas in urban settings. Students at universities encounter many ecological challenges and are equipped to deal with them. For instance, university students frequently deal with a variety of ecological concerns in their everyday lives and studies, such as reducing their carbon footprint, handling garbage on campus, and addressing sustainability issues. These challenges might involve figuring out how social and academic activities affect the environment and figuring out how to fit sustainable practices into their daily routines. Their viewpoints and experiences about green spaces differ, and they can offer new concepts for urban green space development. Universities can provide a significant contribution to a complete knowledge of how students perceive green spaces by combining viewpoints from several disciplines.

It is critical to analyze university students' knowledge, attitudes, and perceptions of urban green areas. This study aimed to investigate the perception of urban green spaces by taking into account the socio-demographic characteristics of university students at Shahjalal University of Science and Technology (SUST), Bangladesh. Understanding university students'

perceptions of urban green spaces can lead to a better understanding of how the younger generation values urban green spaces, which can then be used to mitigate the effects of climate change. Knowing perceptions according to disciplines can also help develop curricula and education not only at university level, but also at primary and secondary levels, where the severity of the consequences of climate change would be significant, and where Bangladesh is one of the most sensitive countries to climate change. Understanding perception can help urban planning by including higher education institutions in recognizing the need for urban green spaces and, consequently, in achieving environmental sustainability.

## 2. Literature review

García Sánchez et al. define urban green spaces as open places with at least 50% permeable surfaces, including parks, gardens, and pedestrian-only zones [13]. Any area that is covered in plants is considered urban green space. All landscaping and "blue spaces" (ponds, lakes, and streams) are included, regardless of size or function. Urban green spaces provide several benefits [4]. It has the potential to improve environmental conditions, biodiversity, outdoor activities, active lifestyles, social contact, and urban health in terms of physical and emotional health. Urban green spaces are places that people perceive could improve their general happiness, economic situation, environment and physical and mental health. The majority of research below describes the extent of urban green spaces and provides some evidence of the variables that influence how people perceive these spaces. The relevant elements that influence how urban green spaces are perceived and their social benefits are discussed in the following section.

### 2.1 Factors influencing the utilization of urban green areas

The usage of urban green areas can be influenced by individual factors such as gender, age, place of residence, level of education, and socioeconomic position. According to research by Jim and Shan, women are more likely than men to appreciate urban green areas in their daily life when visiting Guangzhou. Women are more considerate of urban green spaces and tend to the house. Compared to males, they are more reliant on their surroundings [14]. Pinto et al. find that residents of the urban regions in Portugal, especially those who use stationary parks, showed an interest in the natural world and the surrounding environment. People over 45 used urban green places. The vast majority of users were well educated. Across the country, those who used urban green spaces earned more money on average each month [15]. The value of urban green spaces was closely linked to respondent satisfaction and the cost of residential units [16,17]. The studies suggest that people with higher incomes would have better access to green spaces and report being happier [17]. Furthermore, Marin et al. found notable variations in urban green space according to residential status. Both bigger and smaller villages had different populations. Smaller communities might not have the funding to employ as many professionals in tree and green space management [18]. Haq found that big cities and metropolises have high scores for the urban green factor. Because they have a lot of green area, medium-sized cities also do well on the natural green factor [2]. Urban green space is a public resource that people from different socioeconomic backgrounds should access equitably, although things aren't always this way. High-socioeconomic neighborhoods are more likely to have pleasant walkways, streams, and tree shade [19,20].

Students' main priority when it comes to their ideal settings is parks and other public green spaces. Furthermore, students frequented green spaces and parks for studying, unwinding, and having lunch [21]. For older people compared to younger people, urban green areas are linked to higher levels of well-being and greater aesthetic value. Adults prioritized activities

and mentioned the need for more teen-focused recreation areas where students may report maintenance problems and feel unsafe [22]. The most optimistic people were over 50, and the most pessimistic people were between the ages of 15 and 24. According to Ode Sang et al., respondents with PhDs and professional levels expressed the most optimism on the beneficial benefits of urban green spaces on air quality, whilst those with middle school and lower educational backgrounds expressed the greatest pessimism [23]. However, according to Gearin and Kahle, undergraduate students possess greater awareness than graduates. Students' understanding of urban green places differs depending on their gender. Considering that males are more likely than women to do sports in parks [22]. Additionally, they found that women exercise more than men do in urban green areas. Furthermore, women report feeling more at ease and placing a larger emphasis on the aesthetic aspects of green places [22]. Muslims ought to demand prayer rooms since they pray five times a day. They are limited in their stay by prayer hours. They are unable to pray by going from their homes to green urban spaces. Even in the absence of mosques, park visitors designate a specific area within the parks. They may spend time with their family and engage in religious practice in urban green places [24].

According to a recent survey conducted in Germany by Palliwoda and Priess, elderly people place a higher value on natural components and environmental beauty. On the other hand, younger people prioritized factors such as size, availability, location, and sports facilities [25]. In Spain, older people are less happy with some characteristics of green spaces and more uncomfortable with particular forms of sound [26]. Furthermore, there is a link between age and significant variations in job participation, with retirees having the most unfavorable attitudes toward the characteristics of green areas. Furthermore, it was revealed that people with higher levels of education use the green space for exercise more frequently. Furthermore, people who participate in physical activity in green spaces report being annoyed or affected by noise in the neighborhood less frequently [26]. Ugolini et al. conducted a survey of five European nations (Croatia, Israel, Italy, Lithuania, and Spain) and revealed that urban green space experts often underestimate the worries that users, particularly women, suffer at night. Given the size of the parks, there is a strong correlation between how the public views certain features and how often they use them [27]. Gozalo et al. showed a clear correlation between the amount of green space and the frequency of walking, working out, and unwinding [28]. Cleary et al.'s study found significant positive correlations between middle-aged individuals' perceptions of the quantity of urban green space and their mental health outcomes [29].

## 2.2 The benefits of urban green areas

The presence of green space serves the public in many ways, both directly and indirectly. The main focus of each segment below is the social advantages of urban green space, and the connections between them are emphasized at every turn. Scholars from different disciplines employed quantitative approaches to assess the benefits of urban green spaces and demonstrated the benefits of having access to green space. We will discuss a few of the social benefits of having access to green space in the next section.

**2.2.1 Promoting physical health.** There is a connection between park prescriptions (ParkRx), health issues, and using green spaces [1]. Park prescriptions might help to convey the health advantages of green space. The increased usage of green spaces by middle-aged individuals in Bangladesh may be explained by their poor health and medical professionals' recommendations to exercise and engage in other health-promoting activities outside of the home. There were interior plantations, property gardens, and rooftop plantations on many residences [1]. Islam et al. mentioned that the lack of open space surrounding households contributes to the low amount of vegetation in Bangladesh's border regions. There are real advantages to

gardening for homes. Households claim that they gain financially, environmentally, aesthetically, and recreationally. Households plant a range of fruits in their gardens in order to obtain vitamins and other nutrients. Vegetables and other items cultivated in gardens can satisfy dietary demands [30].

**2.2.2 Psychological wellness.**   Numerous scholarly investigations shown connections between exposure to green areas and a range of positive psychological, emotional, and mental health consequences. For example, Zhou and Rana propose that people can lower their risk of developing conditions such as diabetes, cardiovascular problems, and certain cancers [20]. In addition, children who play in green spaces most of the time have a milder kind of attention deficit disorder (ADD) [31]. According to a Wood et al.'s study, there is a substantial correlation between urban parks' levels of biodiversity and their amenities. The multi-ethnic group's participants—British, Pakistani, Bangladeshi, Indian, and Eastern European—discovered a connection between their experiences of psychological recovery and biodiversity. They consider that increasing the quantity of parks, planting trees to reduce noise pollution in metropolitan areas, and increasing the variety of plants and flowers are all examples of natural therapies with health benefits. Access to green space may not actually improve mental and emotional health since it is difficult to measure the intangible benefits [32].

**2.2.3 Providing economic benefits.**   Landscaping and greenery raise property prices and the financial rewards on land development. Research suggests that financial returns have improved by 5 to 15%, depending on the type of enterprise [33]. Additionally, between 70% and 80% of purchasers stated that natural open space was the most desirable element in a new home complex [34]. Scientists have shown that cities that plant more trees and other greenery save money on cooling expenditures [35]. Cities that have parks, gardens, and public areas benefit economically as well since they offer jobs for construction and maintenance workers and bring in money from visitors [36].

**2.2.4 Enhancing the quality of the environment.**   In addition to offering open wildlife corridors and viewing possibilities, green and open spaces also expand the amount of habitat that is accessible for animals [37]. Because plants absorb carbon dioxide and other elements from the air, urban greening can enhance air quality by reducing pollutants, serving as a noise barrier, lowering urban runoff, and reducing the urban heat island effect [37,38]. Compared to a city block devoid of trees, a roadway lined with trees has around 25% less airborne dust. Even a small number of trees can reduce the amount of dust in the air in heavily crowded regions. Trees on a street provide tiny air currents that help dilute pollutants and lessen the chance of smog and inversions [38].

Based on the debate above, we might say that, particularly in developing countries, scientists, residents, and politicians have started to investigate how urban green space is regarded in recent years. Studies have been done on the subject of whether urban green space affects students from different socioeconomic backgrounds in different ways. Therefore, the current study helps to address the topic of whether or not demographic and disciplinary factors have a greater influence on university students' direct contacts with urban green space. While opinions on urban green space and sociodemographic characteristics may vary, both are crucial in motivating students to lead more environmentally friendly lives. The majority of research on the connection between sociodemographic characteristics and urban green spaces has been conducted in developed nations. Bangladesh's agriculture and livelihoods are greatly impacted by natural catastrophes and climate change, to which it is extremely sensitive. The nation's economy and way of life are so largely reliant on agriculture. There hasn't been much social science study on how people in underdeveloped countries regarding perception urban green space, especially university students in Bangladesh. This study, which is unique in that it examines university students' perceptions of urban green areas, can help future research on the

ways in which the younger generation might contribute to mitigate the adverse impacts of climate change.

## 3. Materials and methods

### 3.1 Research design and study area

An explanatory research design was used for this investigation. Explanatory research can be performed to ascertain why something occurs when information is limited. It can help academics get a deeper grasp of a certain topic, figure out how or why something is happening, or even forecast what will happen in the future [39]. Throughout the inquiry, a quantitative method was utilized by the researchers to examine how students perceive urban green space. Muijs defines quantitative research as the process of obtaining and evaluating numerical data using mathematically based methodologies in order to provide explanations for phenomena [40].

Shahjalal University of Science and Technology (SUST), located in Sylhet, Bangladesh, is the study's site. According to University Grants Commission (UGC), SUST is a public university that receives public funding [41]. There is a substantial tree population all around the SUST. The rich vegetation in this area attracts a lot of tourists since it calms individuals and lessens tension. The researchers' intimate relationships with students, staff, and the academic atmosphere are the main factor in the choice of SUST. There are many of students in this particular area. Being a public university, SUST draws a number of students from beyond the area. Additionally, a diverse spectrum of backgrounds is represented within the university's student body. There are six academic buildings at SUST. IICT Building, A Building, B Building, C Building, D Building, and E Building are the names of the buildings. Samples were chosen from 28 departments spread throughout seven schools in the above indicated buildings.

### 3.2 Population and sampling

There are twenty-seven departments at SUST. Applied Sciences and Technology, Life Sciences, Management and Business Administration, Physical Sciences, Social Sciences, and Agricultural and Mineral Sciences are the six schools that make up SUST. Given that the study aims to provide explanations, student perspectives may provide more accurate information. For the study, students from the following programs were chosen: 2021–22 (1st year), 2020–21 (2nd year), 2019–20 (3rd year), 2018–19 (4th year), and 2021–22 (Master's degree program) because they provided a variety of perspectives on urban green areas. Students' perspectives and comprehension of the subject matter change as they advance through their academic careers. We can obtain a more comprehensive understanding of how attitudes toward urban green areas evolve over time by including students at various stages of their education.

7450 students were enrolled in the academic sessions of 2021–2022 (1st year), 2020–21 (2nd year), 2019–20 (3rd year), 2018–19 (4th year), and 2021–2022 (Master's degree program), according to the office of the register. Cochran's formula was used to calculate 366 samples from 27 departments [42]. The chosen sample size was the population's representative sample size from the normal distribution (Z = 1.96, 95% confidence level). P stands for the necessary proportion. p = 0.5, p+q = 1, q = p-1, and d = 0.05 intended error were the assumptions made by the researcher [42]. We collected data from an extra 72 samples, increasing the dataset despite the 366 sample size. Thus, there are 438 samples in all. As the sample size increases, the results become more accurate. The registrar's office at SUST provides information on the overall number of first-year, second-year, third-year, fourth-year, and master's students involved in this program. Probability sampling is employed for this reason. Simple random sampling is utilized because it provides each participant an equal and fair chance of being picked and since

the student's registration number is known. The final sample is objective and unaffected in its makeup since this selection process gives each participant equal opportunity [43]. Researchers used the program "Research Randomizer" to choose study participants. Every student had an equal chance in this study [44].

## 3.3 Data collection techniques

Participants' information was gathered using the Social Survey Method. The principal research instrument utilized in this study is a questionnaire. A systematic questionnaire was used to collect the data needed for analysis. It consists of a predetermined framework that specifies the precise phrasing and arrangement of a collection of standardized questions [45].

We selected 5 people from the sociology department to take part in data collection. We met with the investigators on several occasions to review the study questionnaires and organize the field visit. We detailed the process of conducting a field study and collecting data from specific participants. We produced 500 questionnaires. We set up two groups. One was made up of students from applied science and technology and physical science, while the other included from other faculties. We wanted to provide questionnaires to a minimum of 10 students in each discipline. Data collection took about 5 months. We began the first phase of data collection from October 5 to 27, 2023. The second phase ran from November 1, 2023 to December 16, 2023. The third phase ran from January 25, 2024 to February 20, 2024.

The research objectives and main topics were presented to the students in class, and questionnaires were distributed for them to complete during class time. Students were given instructions on how to complete the questionnaire and might ask for clarification. They were also informed that the information was private and confidential. Participation was fully optional, and a signed agreement was necessary prior to distributing the questionnaires. The participants were picked using normal random sampling. To eliminate bias, questions were randomly assigned. During data collection, it was unable to control class schedule availability. Not as many students showed up as planned. A few of students appeared uninterested in the given questions and responses. Because of their hectic schedules, it was difficult to get the students to provide the schedule.

**3.3.1 Questionnaires.** A questionnaire was delivered to university students at SUST. The researchers examined several theories offered in published publications, notably those concerning the perception of green spaces as evaluated by Bonaiuto et al. [17]. The researchers then chose the characteristics of perceived urban green areas to include in the questionnaire, and we examined the students' perspectives. The questionnaire is organized into three sections: respondents' sociodemographic information, usage and benefits of urban green spaces, and perceptions of urban green spaces. To help people comprehend the notion, the question is written in both English and Bengali. Because Bengali is our mother language, the students were able to clarify concepts and grasp the topic. And because English is an international language, the abroad students were able to grasp the inquiry.

**3.3.2 Pilot survey.** Prior to the full-scale survey fieldwork, a small-scale pilot survey was carried out. To find out if the respondents comprehended the modified questions, a pilot survey was done. In order to guarantee a thorough representation of all students, a class representative was selected from each of the six university faculties, consisting of second-year undergraduate and master's degree students. Based on their availability to the researcher, participants for this pilot survey were chosen from the following disciplines: business administration, applied science & technology, agricultural & mineral science, social science, physical science, and life science. Twelve students in all answered this survey question. The students were given a thorough explanation of the study's purpose and question by the researchers.

Following the completion of the questionnaire, they offer their opinions on the questions. The researchers modified the items and questions after considering their input.

## 3.4 Measurement and analysis

The satisfaction measures used in this study were taken from a previously validated study by Bonaiuto et al. [17]. The items were an integral part of a comprehensive neighborhood satisfaction survey designed to assess the overall level of satisfaction in a given neighborhood. The study examined in detail people's opinions about the quality of green spaces in different areas, taking into account 8 specific factors (see **Table 1**). The items were assessed using 5-point Likert ratings ranging from 'strongly disagree' to 'strongly agree'. This survey covers both favorable and unfavorable aspects of green spaces, as well as a wide range of viewpoints on their importance. Positively perceived objects refer to how well local green spaces serve as places for leisure and social interaction and provide play opportunities for children. Negative perceived objects refer to the lack or inadequacy of green spaces in residential areas.

The perception of urban green areas is influenced by independent variables such as gender, academic year, disciplinary background, place of residence, most suitable place for living, and home locality vulnerable to climate change (CC) or extreme weather events (EWEs). The perception of urban green areas is the dependent variable. We used crosstab with the Chi-square test, factor analysis and binary logistic regression. A statistical technique called binary logistic regression is used to describe the connection between a categorical independent variable and a dichotomous dependent variable. In social science research, factor analysis is frequently used to reduce the number of variables and identify patterns in the interactions between the variables [46].

## 3.5 Reliability and validity test

Cronbach's Alpha was utilized in this work to evaluate dependability, and component analysis was employed to ascertain the convergent and discriminant validity. According to Cavana et al., the degree to which an instrument reliably and impartially assesses items throughout time and across several instrument sections is known as reliability (error) [47]. A threshold greater than 0.6 in Cronbach's alpha analysis is frequently seen to be appropriate for proving internal consistent dependability [48]. All of the instrument's items had Cronbach's alpha scores that were within the acceptable range of 0.6, indicating the reliability of the tools used in this study (**Table 2**).

Using a factor loading pattern matrix and the component analysis extraction method, the researchers conducted factor analysis on each question to assess the convergent validity of the instrument. A valid instrument, according to Carlson and Herdman, is one where there is a

**Table 1. Items regarding the perception of UGS.**

| Item no | Item |
|---|---|
| Item one | Children are unable to play freely at any park |
| Item two | There are green spaces for relaxing |
| Item three | There are enough green spaces |
| Item four | Green areas are in good condition |
| Item five | Many green areas are disappearing |
| Item six | The green spaces are well-maintained. |
| Item seven | The size of the green areas is insufficient. |
| Item eight | Most green areas are nearby to the public |

Table 2. Summary of reliability and validity test.

| Variable | | No of items | No of sample | Cronbach's Alpha | | |
|---|---|---|---|---|---|---|
| **Perception of Urban Green Space** | Reliability | 8 | 438 | .602 (Fair) | | |
| | | | | Factor Analysis | | |
| | Convergent validity | 8 | 438 | Component | Average Factor Loading | |
| | | | | 3 | .700 | |
| | Discriminant validity | 8 | 438 | | Average Variance Extracted | Component Correlation Matrix |
| | | | | 1 | 0.561 | 0.003 |
| | | | | 2 | 0.399 | 0.003 |
| | | | | 3 | 0.539 | 0.003 |

correlation between the average factor loading of the items that is higher than 0.7 [49]. **Table 2** shows that three components of the eight items pertaining to the perception of urban green space had an average factor higher than.7. The equipment used in the investigation was approved.

The square root of the average variance extracted (AVE) for every component is compared to correlations with other factors in order to assess discriminant validity. According to Fornell and Larcker, discriminant validity is attained when the correlation between the constructs is less than the square root of the average variance extracted (AVE). Three components of the eight items pertaining to the sense of urban green space had an AVE higher than the correlation between the constructs [50]. The statement implies that the instrument used in this study was accepted as legitimate (**Table 2**).

## 4 Result

### 4.1 Sociodemographic characteristics of the participants in the survey

**Table 3** shows that 49.5% of the students were female and 50.5% were male. The majority of the students in the study (45.0%) had a background in social science; the remaining students studied management and business administration (5.3%), life science (6.6%), agricultural and mineral science (6.4%), physical science (17.8%), and applied science and technology (18.9%). First year students made up 31.3% of the surveyed students, fourth year students made up 24.0%, and the remainder students were split up between the Master's degree program (17.6%), second year (16.9%), and third year (10.3%). 38.6% of students live in rural regions, 18.0% in both rural and urban areas, and 43.4% of students live in urban areas. Students were also questioned about their ideal place to reside. The majority of students (50.2%) pick metropolitan regions for their amenities and possibilities, while 49.8% choose rural places for their fresh environment, atmosphere, fresh food, and lower pollution. Students were asked if their locality was vulnerable to climate change and catastrophic weather occurrences. According to their responses, the majority of students' home places (50.5%) are not vulnerable to climate change or extreme weather occurrences, while 49.5% are.

**Table 4** shows that 53.2% of respondents utilize street trees, 18.5% use peace gardens, 13.2% use urban parks, 10.5% use green roofs, and 4.6% use green fields. Additionally, the table reveals that whereas 20.8% of students frequented green spaces 1–3 times each week, 34.0% of respondents said they visited them daily. A little over half of the respondents—57.3% —said they preferred visiting green spaces with friends, while 32.6% said they preferred strolling around them alone. Green spaces are seen by 32.0% of respondents, of whom 28.8% stay for one to two hours, 27.4% stay for less than an hour, and 13.8% stay for longer than two hours.

**Table 3. Socio-demographic information.**

| Variables | Categories | Frequency | Percent |
|---|---|---|---|
| Gender | Female | 217 | 49.5 |
| | Male | 221 | 50.5 |
| | Total | 438 | 100.0 |
| Academic year | 1st year | 137 | 31.3 |
| | 2nd year | 74 | 16.9 |
| | 3rd year | 45 | 10.3 |
| | 4th year | 105 | 24.0 |
| | Master's degree program | 77 | 17.6 |
| | Total | 438 | 100.0 |
| Disciplinary backgrounds | Life science | 29 | 6.6 |
| | Agriculture & Mineral Science | 28 | 6.4 |
| | Applied Science & Technology | 83 | 18.9 |
| | Management & Business Administration | 23 | 5.3 |
| | Physical science | 78 | 17.8 |
| | Social Science | 197 | 45.0 |
| | Total | 438 | 100.0 |
| Place of residence | Rural | 169 | 38.6 |
| | Urban | 190 | 43.4 |
| | Both | 79 | 18.0 |
| | Total | 438 | 100.0 |
| Most suitable area for living | Rural | 218 | 49.8 |
| | Urban | 220 | 50.2 |
| | Total | 438 | 100.0 |
| Home locality vulnerable to CC or EWEs | Yes | 217 | 49.5 |
| | No | 221 | 50.5 |
| | Total | 438 | 100.0 |

**Table 5** shows that 82.6% of respondents generally concur that urban green space improves physical health, whereas 17.4% disagree. 82.6% of the total respondents said they thought urban green space had a good effect on physical health. Of these, 33.3% reported having a high degree of belief, while 42.7% reported having a moderate level. Among the total respondents, 82.6% said that there are chances for physical exercise in urban green areas, and 50.9% particularly noted the physiological benefits of this. Furthermore, as a physiological advantage, urban green areas promote social cohesiveness, according to 12.6% of respondents. The majority of respondents—roughly 87.9%—believe that urban green space positively affects psychological well-being, whereas the remaining participants disagree. 87.9% of the respondents in total said they thought urban green space had a good effect on psychological wellbeing. In particular, 36.3% maintained a moderate conviction in this link, whereas 48.4% firmly believed in it. Of the 87.9% of respondents, 46.6% said that urban green areas are good for psychological health because they reduce stress and improve focus.

Furthermore, according to 34.5% of respondents, urban green spaces have a positive impact on mental health, which is another advantage of these areas. It is estimated that 61.2% of participants believe that urban green areas provide an economic benefit, whereas 38.8% do not think so. 61.2% of the respondents thought that urban green space had some positive economic effects. Of these, 16.0% also had a moderate conviction in the economic advantages of urban green space, whereas 40% of them believed in these benefits. 61.2% of the respondents

**Table 4. Usage of urban green space.**

| Types of urban green spaces | Urban parks | 58 | 13.2 |
|---|---|---|---|
| | Green roofs | 46 | 10.5 |
| | Peace gardens | 81 | 18.5 |
| | Street trees | 233 | 53.2 |
| | Green fields | 20 | 4.6 |
| | Total | 438 | 100.0 |
| Visit or walk through green space | Daily | 149 | 34.0 |
| | 1–3 times per week | 91 | 20.8 |
| | 4–7 times per week | 89 | 20.3 |
| | Few times a month | 43 | 9.8 |
| | Monthly or less | 17 | 3.9 |
| | Occasionally | 49 | 11.2 |
| | Total | 438 | 100.0 |
| Usually go with green space | Alone | 143 | 32.6 |
| | Friends | 251 | 57.3 |
| | Partners | 21 | 4.8 |
| | Family members | 23 | 5.3 |
| | Total | 438 | 100.0 |
| Staying in green space | Just pass-through | 140 | 32.0 |
| | Less than 1 hour | 120 | 27.4 |
| | 1–2 hours | 126 | 28.8 |
| | More than 2 hours | 54 | 13.8 |
| | Total | 438 | 100.0 |

overall said that non-timber forest products from urban green space have a positive economic impact. Furthermore, as an added financial benefit, 37.7% of respondents stated that urban green space raises property prices. Of the participants, about 85.4% think that urban green areas improve the quality of the environment, while 14.6% disagree. Of those surveyed, 85.4% had a strong conviction in the positive environmental effects of urban green space, whilst 31.3% held a moderate belief in the same advantages. Of the total respondents, 47.0% saw this as a positive for the environment and 85.4% agreed that urban green areas help reduce carbon dioxide emissions. Furthermore, urban green areas were recognized by 24.2% of participants as a safeguard against natural hazards and as an environmentally beneficial feature.

## 4.2 Perception of urban green spaces

The aim of this study was to assess the impact of the socio-demographic composition of students at SUST on their perceptions of urban green spaces. First, we investigated the relationship between sociodemographic characteristics and the benefits of urban green spaces. Next, binary logistic regression was used to distinguish between participants with a favourable or negative perception of urban green spaces. Factor analysis was used to assess the appropriateness of sampling items related to the perception of urban green spaces.

**4.2.1 Association between socio-demographic factors and the benefits of urban green spaces.** The study employed Chi-Square Tests (Turney, 2022) to examine the association between students' perceptions of urban green spaces and sociodemographic characteristics. This study's dependent variable is the benefits of urban green spaces, including their positive effects on the physical health, psychological health, economy and environment. The variables in this study include gender (Male = 0, Female = 1), academic year (1st year = 1, 2nd year = 2,

**Table 5. Benefits of urban green space.**

| | | | |
|---|---|---|---|
| Physical health benefits | Yes | 362 | 82.6 |
| | No | 76 | 17.4 |
| | Total | 438 | 100.0 |
| Magnitude of physical health benefits | Low | 29 | 6.6 |
| | Moderate | 187 | 42.7 |
| | High | 146 | 33.3 |
| | Total | 362 | 82.6 |
| Physiological benefits for promoting physical health | Provide space for physical activity | 223 | 50.9 |
| | Promote social cohesion | 55 | 12.6 |
| | Increase life expectancy | 51 | 11.6 |
| | Relief from tiredness | 33 | 7.5 |
| | Total | 362 | 82.6 |
| Psychological health benefits | Yes | 385 | 87.9 |
| | No | 53 | 12.1 |
| | Total | 438 | 100.0 |
| Magnitude of psychological health benefits | Low | 16 | 3.7 |
| | Moderate | 159 | 36.3 |
| | High | 212 | 48.4 |
| | Total | 387 | 88.4 |
| Psychological benefits of urban green spaces | Improve mental well-being | 151 | 34.5 |
| | Relieve stress and attention | 204 | 46.6 |
| | Decrease aggressive behaviour | 24 | 5.5 |
| | Reduces depression | 8 | 1.8 |
| | Improve mental well-being | 387 | 88.4 |
| Economic benefits | Yes | 268 | 61.2 |
| | No | 170 | 38.8 |
| | Total | 438 | 100.0 |
| Magnitude of economic benefits | Low | 27 | 6.2 |
| | Moderate | 175 | 40.0 |
| | High | 70 | 16.0 |
| | Total | 272 | 62.1 |
| Beneficial for providing economic benefits | Raise property values | 59 | 13.5 |
| | Provides non-timber forests products | 165 | 37.7 |
| | Decrease water treatment costs | 30 | 6.8 |
| | Creates new parks for green vision | 18 | 4.1 |
| | Total | 272 | 62.1 |
| Environmental benefits | Yes | 374 | 85.4 |
| | No | 64 | 14.6 |
| | Total | 438 | 100.0 |
| Magnitude of environmental benefits | Low | 12 | 2.7 |
| | Moderate | 137 | 31.3 |
| | High | 232 | 53.0 |
| | Total | 381 | 87.0 |
| Benefit for improving environmental quality | Good place for animal sanctuary and fresh air | 50 | 11.4 |
| | Safety from natural hazards | 106 | 24.2 |
| | Reduce carbon dioxide emission | 206 | 47.0 |
| | Reduces soil erosion in nearby areas | 19 | 4.3 |
| | Total | 381 | 87.0 |

3rd year = 3; 4th year = 4, Master's degree program = 5), disciplinary backgrounds (life science = 1, Agricultural and Mineral Science = 2, Applied Science and Technology = 3, Management & Business administration = 4, Physical Science = 5, Social Science = 6), Place of residence (Both = 2, Urban = 1, Rural = 0), suitable living place (Urban = 1, Rural = 0), and vulnerability to extreme weather occurrences or climate change in an individual's home area (yes = 1, No = 0). The dependent variables consist of perceived benefits of urban green spaces, which include physical health benefits (yes = 1, no/don't know = 0), psychological health benefits (yes = 1, no/don't know = 0), economic advantages (yes = 1, no/don't know = 0), and environmental benefits (yes = 1, no/don't know = 0).

*4.2.1.1 Physical health benefits.* Table 6 shows that three variables—gender, study year, and home locality's susceptibility to extreme weather events and climate change—have a statistically significant link with the physical health benefits of urban green space. Additionally, it demonstrates that 60.0% of individuals who did not experience extreme weather or climate change in their local communities agreed that the benefits to physical health were worthwhile. According to the survey, 86.9% of participants who experienced harsh weather in their area agreed that urban green areas are beneficial for physical health. It has been shown that, at a 5% significance level, there is a statistically significant correlation between the benefits of physical health and one's home location (p = .018). While second-year students (75.7%) believe that urban green areas are beneficial to physical health, the majority of master's program students (92.2%) do. At the 5% level of significance, there is a statistically significant association (p =

**Table 6. Socio-demographic determinants and physical health benefits of UGC.**

| Variables | Physical health benefit | | Total | Sig |
|---|---|---|---|---|
| | Yes | No | | |
| **Gender** | | | | **.000** |
| Male | 163 (73.8%) | 58 (26.2%) | 221 (100.0%) | |
| Female | 199 (91.7%) | 18 (8.3%) | 217 (100.0%) | |
| **Academic year** | | | | **.024** |
| 1st year | 106 (77.4%) | 31 (22.6%) | 137 (100.0%) | |
| 2nd year | 56 (75.7%) | 18 (24.3%) | 74 (100.0%) | |
| 3rd year | 39 (86.7%) | 6 (13.3%) | 45 (100.0%) | |
| 4th year | 90 (85.7%) | 15 (14.3%) | 105 (100.0%) | |
| Master's degree program | 71 (92.2%) | 6 (7.8%) | 77 (100.0%) | |
| **Disciplinary background** | | | | .253 |
| Life science | 27(93.1%) | 2 (6.9%) | 29 (100.0%) | |
| Agriculture & Mineral Science | 24 (85.7%) | 4 (14.3%) | 28 (100.0%) | |
| Applied Science & Technology | 70 (84.3%) | 13 (15.7%) | 83 (100.0%) | |
| Management & Business Administration | 20 (87.0%) | 3 (13.0%) | 23 (100.0%) | |
| Physical science | 58 (74.4%) | 20 (25.6%) | 78 (100.0%) | |
| Social Science | 163 (82.7%) | 34 (17.3%) | 197 (100.0%) | |
| **Place of residence** | | | | .267 |
| Rural | 134 (79.3%) | 35 (20.7%) | 169 (100.0%) | |
| Urban | 163 (85.8%) | 27 (14.2%) | 190 (100.0%) | |
| Both | 65 (82.3%) | 14 (17.7%) | 79 (100.0%) | |
| **Most suitable place for living** | | | | .119 |
| Urban | 188 (85.5%) | 32 (14.5%) | 220 (100.0%) | |
| Rural | 174 (79.8%) | 44 (20.2%) | 218 (100.0%) | |
| **Home locality vulnerable to CC or EWEs** | | | | **.018** |
| yes | 170 (78.3%) | 47 (21.7%) | 217 (100.0%) | |
| no | 192 (86.9%) | 29 (13.1%) | 221 (100.0%) | |

*p-value < 0.1

**p-value < 0.05

***p-value < 0.01.

.024) between the number of years spent studying and the beneficial impacts on physical health. The majority of female students perceived urban green spaces as beneficial to physical health (91.7%), in contrast to male students (73.8%). The chi-square test shows that variables such as disciplinary background, place of residence and suitable living space were not statistically significant. Also, the results suggest that the physical health benefits of urban green spaces are not influenced by, or dependent on, specific characteristics.

*4.2.1.2 Psychological well-being.* Connected to the psychological well-being of parks and other urban green space are socio-demographic variables, as shown in **Table 7.** Table data reveals a statistically significant relationship between gender, academic year, and home location's susceptibility to climate change and severe weather occurrences and the psychological well-being of urban green space. Those who did not live in regions affected by climate change or severe weather reported higher rates of psychological well-being (90.5%). According to the table, 85.3% of people who lived in areas hit by severe weather agreed that parks and other urban green space improve people's psychological well-being. Although a lesser number of second-year students hold the notion that urban green areas increase psychological well-being (81.1%), the majority of master's degree students (96.1%), agree. Among university students, females are more likely to agree than males that access to urban green space improves mental health (90.8% vs. 85.1%). Discipline background, place of residence, and permissible living conditions were not determined to be statistically significant variables according to the Chi-Square test. These characteristics do not seem to affect the psychological well-being of urban green areas, according to the study's results.

Table 7. Socio-demographic determinants and psychological well-being of UGC.

| Variables | Psychological health benefits | | Total | Sig |
|---|---|---|---|---|
| | Yes | No | | |
| **Gender** | | | | **.067** |
| Male | 188 (85.1%) | 33 (14.9%) | 221 (100.0%) | |
| Female | 197 (90.8%) | 20 (9.2%) | 217 (100.0%) | |
| **Academic year** | | | | **.052** |
| 1st year | 117 (85.4%) | 20 (14.6%) | 137 (100.0%) | |
| 2nd year | 60 (81.1%) | 14 (18.9%) | 74 (100.0%) | |
| 3rd year | 41 (91.1%) | 4 (8.9%) | 45 (100.0%) | |
| 4th year | 93 (88.6%) | 12 (11.4%) | 105 (100.0%) | |
| Master's degree program | 74 (96.1%) | 3 (3.9%) | 77 (100.0%) | |
| **Disciplinary background** | | | | .369 |
| Life science | 25 (86.2%) | 4 (13.8%) | 29 (100.0%) | |
| Agriculture & Mineral Science | 26 (92.9%) | 2 (7.1%) | 28 (100.0%) | |
| Applied Science & Technology | 77 (92.8%) | 6 (7.2%) | 83 (100.0%) | |
| Management & Business Administration | 21 (91.3%) | 2 (8.7%) | 23 (100.0%) | |
| Physical science | 64 (82.1%) | 14 (17.9%) | 78 (100.0%) | |
| Social Science | 172 (87.3%) | 25 (12.7%) | 197 (100.0%) | |
| **Place of residence** | | | | .442 |
| Rural | 147 (87.0%) | 22 (13.0%) | 169 (100.0%) | |
| Urban | 171 (90.0%) | 19 (10.0%) | 190 (100.0%) | |
| Both | 67 (84.8%) | 12 (15.2%) | 79 (100.0%) | |
| **Most suitable place for living** | | | | .100 |
| Urban | 199 (90.5%) | 21 (9.5%) | 220 (100.0%) | |
| Rural | 186 (85.3%) | 32 (14.7%) | 218 (100.0%) | |
| **Home locality vulnerable to CC or EWEs** | | | | **.092** |
| yes | 185 (85.3%) | 32 (14.7%) | 217 (100.0%) | |
| no | 200 (90.5%) | 21 (9.5%) | 221 (100.0%) | |

*p-value < 0.1

**p-value < 0.05

***p-value < 0.01.

**Table 8. Socio-demographic determinants and economic benefits of UGC.**

| Variables | Economic benefits | | Total | Sig |
|---|---|---|---|---|
| | Yes | No | | |
| **Gender** | | | | .070 |
| Male | 126 (57.0%) | 95 (43.0%) | 221 (100.0%) | |
| Female | 142 (65.4%) | 75 (34.6%) | 217 (100.0%) | |
| **Academic year** | | | | .657 |
| 1st year | 84 (61.3%) | 53 (38.7%) | 137 (100.0%) | |
| 2nd year | 41 (55.4%) | 33 (44.6%) | 74 (100.0%) | |
| 3rd year | 28 (62.2%) | 17 (37.8%) | 45 (100.0%) | |
| 4th year | 63 (60.0%) | 42 (40.0%) | 105 (100.0%) | |
| Master's degree program | 52 (67.5%) | 25 (32.5%) | 77 (100.0%) | |
| **Disciplinary background** | | | | .863 |
| Life science | 16 (55.2%) | 13 (44.8%) | 29 (100.0%) | |
| Agriculture & Mineral Science | 20 (71.4%) | 8 (28.6%) | 28 (100.0%) | |
| Applied Science & Technology | 50 (60.2%) | 33 939.8%) | 83 (100.0%) | |
| Management & Business Administration | 15 (65.2%) | 8 (34.8%) | 23 (100.0%) | |
| Physical science | 47 (60.3%) | 31 (39.7%) | 78 (100.0%) | |
| Social Science | 120 (60.9%) | 77 (39.1%) | 197 (100.0%) | |
| **Place of residence** | | | | .767 |
| Rural | 107 (63.3%) | 62 (36.7%) | 169 (100.0%) | |
| Urban | 114 (60.0%) | 76 (40.0%) | 190 (100.0%) | |
| Both | 47 (59.5%) | 32 (40.5%) | 79 (100.0%) | |
| **Most suitable place for living** | | | | .752 |
| Urban | 133 (60.5%) | 87 (39.5%) | 220 (100.0%) | |
| Rural | 135 (61.9%) | 83 (38.1%) | 218 (100.0%) | |
| **Home locality vulnerable to CC or EWEs** | | | | .527 |
| yes | 136 (62.7%) | 81 (37.3%) | 217 (100.0%) | |
| no | 132 (59.7%) | 89 (40.3%) | 221 (100.0%) | |

*p-value < 0.1

**p-value < 0.05

***p-value < 0.01.

*4.2.1.3 Economic benefits.* The relationship between socio-demographic characteristics and the financial benefits of urban green spaces is presented in **Table 8**. The information in the table indicates that there is a statistically significant correlation between gender and the financial benefits of urban green spaces. A p-value of .070 indicates a statistically significant association between gender and economic benefits at 10% level. The test results show that female students are more likely than male students to believe that urban green spaces offer economic benefits (65.4%) than (57.0%). Characteristics such as discipline, academic year, place of residence, suitable living environment and sensitivity to extreme weather or climate change in the region of origin were not statistically significant, according to chi-square tests. The results imply that these factors neither influence nor determine the financial benefits of urban green spaces.

*4.2.1.4 Environmental benefits.* **Table 9** summarizes the environmental benefits of urban green space as well as the interaction of socio-demographic parameters. The table reveals that four variables—gender, academic year, disciplinary backgrounds, and hometown sensitivity to climate change and severe weather—are strongly associated to the good impacts of urban green areas on environmental health. The Crosstab result reveals that 90.5% of participants who had not experienced extreme weather or climate change in their individual localities showed support for environmental benefits. 80.2% of respondents who reported witnessing extreme weather events in their individual localities agreed that urban green spaces provide an environmental advantage. At a statistical significance level of 5% (p = .002), the reported

**Table 9. Socio-demographic determinants and environmental benefits of UGC.**

| Variables | Environment benefits | | Total | Sig |
|---|---|---|---|---|
| | Yes | No | | |
| **Gender** | | | | .000 |
| Male | 172 (77.8%) | 49 (22.2%) | 221 (100.0%) | |
| Female | 202 (93.1%) | 15 (6.9%) | 217 (100.0%) | |
| **Academic year** | | | | .005 |
| 1st year | 109 (79.6%) | 28 (20.4%) | 137 (100.0%0 | |
| 2nd year | 59 (79.7%) | 15 (20.3%) | 74 (100.0%) | |
| 3rd year | 42 (93.3%) | 3 (6.7%) | 45 (100.0%) | |
| 4th year | 90 (85.7%) | 15 (14.3%) | 105 (100.0%) | |
| Master's degree program | 74 (96.1%) | 3 (3.9%) | 77 (100.0%) | |
| **Disciplinary background** | | | | .058 |
| Life science | 24 (82.8%) | 5 (17.2%) | 29 (100.0%) | |
| Agriculture & Mineral Science | 26 (92.9%) | 2 (7.1%) | 28 (100.0%) | |
| Applied Science & Technology | 73 (88.0%) | 10 (12.0%) | 83 (100.0%) | |
| Management & Business Administration | 21 (91.3%) | 2 (8.7%) | 23 (100.0%) | |
| Physical science | 58 (74.4%) | 20 (25.6%) | 78 (100.0%) | |
| Social Science | 172 (87.3%) | 25 (12.7%) | 197 (100.0%) | |
| **Place of residence** | | | | .261 |
| Rural | 148 (87.6%) | 21 (12.4%) | 169 (100.0%) | |
| Urban | 163 (85.8%) | 27 (14.2%) | 190 (100.0%) | |
| Both | 63 (79.7%) | 16 (20.3%) | 79 (100.0%) | |
| **Most suitable place for living** | | | | .395 |
| Urban | 191 (86.8%) | 29 (13.2%) | 220 (100.0%) | |
| Rural | 183 (83.9%) | 35 (16.1%) | 218 (100.0%) | |
| **Home locality vulnerable to CC or EWEs** | | | | .002 |
| yes | 174 (80.2%) | 43 (19.8%) | 217 (100.0%) | |
| no | 200 (90.5%) | 21 (9.5%) | 221 (100.0%) | |

*p-value < 0.1

**p-value < 0.05

***p-value < 0.01.

relationship between environmental advantages and vulnerability to home locale is significant. Students enrolled in master's degree programs are more likely than first-year students to feel that urban green areas enhance environmental advantages (96.1%), with 79.6% of first-year students holding this belief. The observed correlation between study years and environmental advantages is statistically significant at the 5% level (p = .005).

There is a strong statistical relationship between gender and environmental benefits (p = .000). According to the test, female respondents (93.1%) are more likely than male students (77.8%) to feel that urban green areas give environmental advantages. Our study found that a higher percentage (92.9%) of students in agriculture and mineral sciences supported the concept of environmental benefits than 91.3% of students in management & business administration. The observed link between academic fields and the environmental advantages of urban green space is statistically significant at the 10% level (p = .058). Nevertheless, the Chi-Square test shows that factors including housing location and appropriate living arrangement did not show statistical significance. The study's results support the idea that these factors neither impact nor determine the environmental advantages of urban green areas.

**4.2.2 Relationship between socio-demographic factors and perceptions of urban green spaces: Binary logistic regression.** The variables considered for binary logistic regression in this study are: gender (male = 0, female = 1); academic year (1st year = 1, 2nd year = 2, 3rd year = 3; 4th year = 4, master's program = 5); disciplines (life sciences = 1, agricultural and mineral sciences = 2, applied sciences and technology = 3, management and business

administration = 4, physical sciences = 5, social sciences = 6); place of residence (both = 2, urban = 1, rural = 0); living in an urban area (under 15 years = 1; 15–20 years = 2; over 20 years = 3); living in a rural area (under 15 years = 1; 15–20 years = 2; over 20 years = 3); and sensitivity to extreme weather events or climate change in the individual's home region (yes = 1, no = 0). The dependent variable is the item relating to the perception of urban green spaces. In this case, we calculated the sum of the items relating to the perception of urban green spaces. We then calculated the mean of the "perception of urban green spaces" variable. The average is 19. We classified the "perception of urban green spaces" variable into two categories: high and low. The range from 8 to 18 is coded 0 and the range from 19 to 40 is coded 1.

When all factors were included in the Binary Logistic Regression model, the findings were statistically significant (p < .005), suggesting that the model can distinguish between respondents with high and low perceptions of urban green areas. The model accurately identified 74.7% of urban green space perceptions, accounting for 34.1% (Cox and Snell $R^2$) and 45.4% (Nagelkerke $R^2$) of the observations (**Table 10**). Table 10 demonstrates that the only two independent factors that significantly contributed to the model were residing in urban regions and

**Table 10. Parameter estimates for levels associated with perception of urban green spaces.**

| Variables | Coefficient | Sig. | Odds Ratio | 95% C.I. for Odds Ratio | |
|---|---|---|---|---|---|
| | | | | Lower | Upper |
| **Gender** | | | | | |
| Female (1) | -.573 | .332 | .564 | .177 | 1.795 |
| **Ref** (Male 0) | | | | | |
| **Academic year** | | .131 | | | |
| 2nd year (2) | -1.761 | .077 | .172 | .024 | 1.214 |
| 3rd year (3) | -.523 | .626 | .593 | .072 | 4.856 |
| 4th year (4) | -2.006 | **.018** | .134 | .026 | .705 |
| Masters (5) | -1.675 | .073 | .187 | .030 | 1.168 |
| **Ref** (1st year 1) | | | | | |
| **Disciplinary background** | | .945 | | | |
| Agriculture and Mineral Science (2) | -.641 | .714 | .527 | .017 | 16.299 |
| Applied Science and Technology (3) | .016 | .988 | 1.016 | .112 | 9.262 |
| Management and business administration (4) | 21.576 | .999 | 2346570705.306 | .000 | . |
| Physical science (5) | .536 | .652 | 1.709 | .167 | 17.518 |
| Social science (6) | .650 | .549 | 1.916 | .228 | 16.111 |
| **Ref** (Life Science 1) | | | | | |
| **Place of residence** | | 1.000 | | | |
| Urban (2) | -40.802 | .999 | .000 | .000 | . |
| Both (3) | -19.340 | .999 | .000 | .000 | . |
| **Ref** (Rural 1) | | | | | |
| **Living rural area** | | .696 | | | |
| 15–20 year (2) | .501 | .395 | 1.651 | .520 | 5.234 |
| More than 20 years (3) | -40.778 | .998 | .000 | .000 | . |
| **Ref** (Less than 15 years 1) | | | | | |
| **Living urban area** | | **.025** | | | |
| 15–20 year (2) | .315 | .644 | 1.370 | .361 | 5.205 |
| More than 20 years (3) | -3.485 | **.009** | .031 | .002 | .419 |
| **Ref** (Less than 15 years 1) | | | | | |
| **Home locality vulnerable to CC or EWEs** | | | | | |
| Yes (1) | 1.102 | **.069** | 3.010 | .918 | 9.863 |
| **Ref** (No 0) | | | | | |

**Chi-square** = 34.561; **R² (Cox & Snell)** = 34.1%; **R² (Nagelkerke)** = 45.4%; **Classification** = 74.7%; **Sig** = .007

*p-value < 0.1

**p-value < 0.05

***p-value < 0.01.

being susceptible to one's home neighborhood. The most important indicators were being an urban resident. Students who have lived in urban regions for more than 20 years are likely to see urban green spaces less favorably than those who have lived there for less than 15 years, as shown by their likelihood of having .031.

Not statistically significant at the 5% level were the other explanatory factors, which included gender, academic year, disciplinary backgrounds, place of residence, living in rural regions, and susceptibility to home location connected to EWEs and climate change. At a 10% significance level, there was a correlation found between the sense of urban green areas and susceptibility to residential location due to EWEs and climate change. The odds ratio indicated that individuals without personal experience with extreme weather events or climate change were 3.01 times more likely to have a positive opinion of urban green spaces than people with personal experience with these phenomena.

**4.2.3 Factor analysis & Varimax Rotation Matrix for the perception of urban green spaces.** The Kaiser-Meyer-Olkin (KMO) measure of sample adequacy for the items relating to perceptions of urban green spaces was 0.660, according to the factor analysis results. For these eight items, a KMO score greater than 0.6 was used to determine their factorability. Table 11 displays the three components that the research found to have Eigen values greater than 1.0. The Varimax rotation approach with a 0.50 correlation coefficient cutoff is used in the investigation. From Table 11, it was observed that a total of three items presented significant loadings on the first factor. These items are as follows: There are enough green spaces; Green spaces are in good condition; Green spaces are well equipped. Factor two comprises three items. These are as follows: There are green spaces to relax in; Many green spaces are disappearing; The green spaces are too small. Factor three comprises two items: There are no parks where children can play freely; Most green spaces are closed to the public.

## 5 Discussion

The present study examines university students' perceptions of urban green spaces, and how sociodemographic and disciplinary factors shape these perceptions. Using the chi-square test

**Table 11. Factor analysis for perception of urban green spaces.**

|  | Component | | |
|---|---|---|---|
|  | 1 | 2 | 3 |
| **Factor 1** |  |  |  |
| There are enough green areas | .802 |  |  |
| Green areas are in good condition | .802 |  |  |
| The green areas are well-equipped | .565 |  |  |
| **Factor 2** |  |  |  |
| There are green areas for relaxing |  | -.643 |  |
| Many green areas are disappearing |  | .611 |  |
| The green areas are too small |  | .652 |  |
| **Factor 3** |  |  |  |
| There is no park where children can play freely |  |  | .760 |
| Most green areas are closed to the public |  |  | .663 |

**Eigenvalue**
Component 1 = 2.180; Component 2 = 1.325; Component 3 = 1.014

**Percentage of variance explained (%)**
Component 1 = 27.244; Component 2 = 16.568; Component 3 = 12.671

**Kasier-Meyer-Olkin** = .660

**Bartletts' test of spericity approx. chi square** = 367.104; **df** = 28; **Sig** = .000

and binary logistic regression analysis to investigate the correlation between students' socio-demographic characteristics and their understanding and view of urban green spaces, the study found a significant correlation between gender, academic year, disciplinary backgrounds, urban residence, hometown sensitivity to extreme weather events or climate change, and perception of urban green spaces. Female students have a better understanding and perception of urban green spaces than male students. The results are consistent with other research. For example, Speake et al. found a gender gap in students' understanding of urban green spaces. Adults focused on activities and mentioned the need for additional recreational parks for teens, while students noted maintenance issues and worried about their safety [21]. Gearin and Kahle found that women engage in more physical activity in urban green spaces than men in Los Angeles. Women also appreciate the visual appeal of green spaces and experience a greater sense of well-being in these environments [22]. Jim and Shan found that women in Guangzhou attach more importance to urban green spaces than men. This heightened sensitivity is attributed to women's role in household management, which makes them more dependent on and aware of their environment [14]. According to the present study, older students are less likely than younger students to embrace urban green space. The results are not the same as those of Ode Sang et al., who found that those with PhDs and professional backgrounds had the most favorable opinions on urban green spaces' ability to improve air quality, whereas people with only a middle school degree or less had the most negative opinions [23].

This study shows that there is no statistical relationship between students' previous place of residence and their opinion of urban green space. Students who have lived in urban areas for more than 20 years are more likely to evaluate urban green space negatively. The results are consistent with the findings of previous research. Jim and Shan found no significant differences in perceptions of urban green space based on residents' location [14]. The study indicates that the benefits of urban green space, including physiological, psychological and economic benefits, are closely related to factors such as gender, academic year and vulnerability to the local environment. Urban living was identified as the most influential factor through binary logistic regression analysis. Students who lived in urban areas for more than 20 years had lower perceptions of urban green space. This finding confirms the hypothesis of the study. Jim and Shan found significant differences in perception based on socioeconomic characteristics such as gender, education and place of residence [14].

## 6 Concluding remarks and recommendation

Any city's layout is largely influenced by its parks and other open areas. Bangladesh's unplanned infrastructure, poverty, and rapid development make managing green spaces challenging and costly. Green space management is not just desired but also required to preserve urban sustainability in a country like Bangladesh that is experiencing climate change. It was discovered that students enjoy urban green areas. Based on factors including gender, academic year, disciplinary backgrounds, place of residence, and how vulnerable their hometown is to extreme weather or climate change, our survey revealed that majority of the students were very appreciative of urban green areas. Planning and regulations for managing urban green spaces should take these differences across green areas into consideration. Enough assurances should be provided by green areas for a range of applications centered on enjoyment and beauty. Simultaneously, planners should include social and recreational activities into daily-use green areas. Understanding various perspectives on urban parks is essential if we are to take proper urban management seriously going forward.

This research made a compelling case for the need for sufficient planting space in Bangladesh's cities. In the country, local and national authorities must work together to manage

urban green areas, with master plans and zoning restrictions serving as a means of monitoring planning. Therefore, it is imperative that local non-governmental organizations (NGOs) and community organizations take the lead in advocating for the creation of green spaces and ensuring that community concerns are taken into account during the planning process. We also need to participate in the technical aspect of planting trees. It is risky to grow trees on the rooftops of old buildings. If the roof becomes wet, accidents may happen. On the other hand, inexpensive chemicals are being used to protect roofs in newly built structures. In addition, officials and researchers need to work together rather than scattering or disorganized tree planting. Working together with researchers generates innovative ideas and improves outcomes.

## 7 Limitations and future research

The basic sampling procedure of our study was challenging to carry out due to time constraints, academic semester exams, winter vacations, political unpredictability and class schedule issues. To get more detailed results, further research on different topics is needed to explore students' perspectives on urban green space from other universities in Bangladesh and if possible, transnational comparative research, such as to South Asian countries, can be conducted.

Participants' opinions about urban green space were not affected by their place of residence during the one-year study. The result might be different after three to five years. The authors urged more research over a longer period of time. Most research on the effectiveness of urban parks in combating climate change has focused on affluent countries, even though developing countries are more susceptible. Future research should focus on identifying areas with urban green space that can help reduce the impacts of climate change, with a focus on examining outcomes rather than just numbers. The findings of the research can also provide information so that the developed world can learn more about the effects of urban green space on reducing greenhouse gas emissions by conducting large-scale research. Developed and underdeveloped countries may come to different conclusions. This work can be useful to social and environmental scientists and forestry researchers, urban planners who want to develop new training seminars or incorporate data-validated models into existing reform initiatives.

## Supporting information

**S1 Dataset. Data set used for analysis and results presented in the study.**
(SAV)

## Author Contributions

**Conceptualization:** Shah Md Atiqul Haq.

**Data curation:** Bijoya Saha, Shah Md Atiqul Haq.

**Formal analysis:** Bijoya Saha, Shah Md Atiqul Haq.

**Investigation:** Bijoya Saha, Shah Md Atiqul Haq.

**Methodology:** Bijoya Saha, Shah Md Atiqul Haq.

**Software:** Bijoya Saha.

**Supervision:** Shah Md Atiqul Haq.

**Validation:** Bijoya Saha, Shah Md Atiqul Haq.

**Visualization:** Bijoya Saha, Shah Md Atiqul Haq.

**Writing – original draft:** Bijoya Saha, Shah Md Atiqul Haq.

**Writing – review & editing:** Bijoya Saha, Shah Md Atiqul Haq.

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
