## [Decision Letter · Decision Letter 0]

26 Jun 2024

PONE-D-24-18205Perception of urban green space among university students in BangladeshPLOS ONE

Dear Dr. Atiqul Haq,

Thank you for submitting your manuscript to PLOS ONE. After careful consideration, we feel that it has merit but does not fully meet PLOS ONE’s publication criteria as it currently stands. Therefore, we invite you to submit a revised version of the manuscript that addresses the points raised during the review process.Please ensure that your decision is justified on PLOS ONE’s publication criteria and not, for example, on novelty or perceived impact.

We look forward to receiving your revised manuscript.

Kind regards,

Gouranga Lal Dasvarma, PhD

Academic Editor

PLOS ONE

Journal Requirements:

2. In the online submission form, you indicated that this is a survey study conducted by the authors. If any party requests for data, it can be available to them but not publicly available.

3. We note that Figure 3 in your submission contain [map/satellite] images which may be copyrighted. All PLOS content is published under the Creative Commons Attribution License (CC BY 4.0), which means that the manuscript, images, and Supporting Information files will be freely available online, and any third party is permitted to access, download, copy, distribute, and use these materials in any way, even commercially, with proper attribution. For these reasons, we cannot publish previously copyrighted maps or satellite images created using proprietary data, such as Google software (Google Maps, Street View, and Earth). For more information, see our copyright guidelines: http://journals.plos.org/plosone/s/licenses-and-copyright.

a. You may seek permission from the original copyright holder of Figure 3 to publish the content specifically under the CC BY 4.0 license.  

Additional Editor Comments:

Please address the comments of the two reviewers thoroughly and satisfactorily and have the manuscript edited by a professional English editor.

Reviewers' comments:

Reviewer's Responses to Questions

**Comments to the Author**

1. Is the manuscript technically sound, and do the data support the conclusions?

Reviewer #1: Partly

Reviewer #2: No

2. Has the statistical analysis been performed appropriately and rigorously? 

Reviewer #1: Yes

Reviewer #2: Yes

3. Have the authors made all data underlying the findings in their manuscript fully available?

Reviewer #1: Yes

Reviewer #2: No

4. Is the manuscript presented in an intelligible fashion and written in standard English?

Reviewer #1: No

Reviewer #2: No

5. Review Comments to the Author

Reviewer #1: This paper offers an important and timely contribution on the perceptions of urban green space from a developing nation's perspective. I commend the author's choice in surveying university students as younger generations are increasingly more cognisant of environmental issues and cues, and, as the author noted, are not always involved in decision-making processes.

While the paper has a great deal of potential and provides a much needed perspective from a developing country so acutely impacted by climate change and rapid urbanisation, the paper falls short in a number of areas. First, the literature review does not provide appropriate context or definitions of typologies of green space and how this is defined in a Bangladeshi context (in Section 3.1). There was also scope to strengthen the justification for surveying university students. Please see attached document for additional recommended literature.

The methodology section can be improved by justifying the quantitative approach. When investigating 'perceptions', why weren't qualitative methods such as interviews or focus groups employed to provide a more nuanced perspective? Additionally, the author states that the survey design is based on the work by Bonaiuto et al. (2003). However, there is no indication of what these perceptions are nor is this work mentioned in the Literature Review.

Notwithstanding that the author may be from a NESB, there are several instances where meaning is lost due to grammatical errors and issues in written expression (refer to the attached document). Several assertions also require clarification. For example, in how does one know whether it’s not the wealth that leads to happiness? (Line 102) and why do university students have a greater understanding of UGS?

Returning to context, the paper could be further strengthened with the inclusion of a short discussion on the context of green space planning in Bangladesh. For instance, how are UGS governed and are communities consulted in decision-making processes? This would be particularly beneficial to form a more robust conclusion and set of recommendations.

In summary, this paper has the potential to offer critical insights into perceptions of UGS from the developing South. However, it falls short in several areas, and thus requires major revision. I recommend that the author enhance the literature review, methodology, and English language quality and provide stronger justification for the adoption of a quantitative methodology as opposed to qualitative or mixed-methods design.

Reviewer #2: Authors have analyzed perception of university students for UGS however manuscript needs significant rewriting. Flow of writing is broken to that extent that not even para , each sentence is disjointed with previous sentence that makes reading not enjoyable.

Besides, Introduction and Literature review is written in the form of report rather than a research paper. In fact, Introduction fails to establish the need of the study. I suggest Authors to mix writeup from discussion section in Introduction to clearly bring out the need of the study. The Authors need to answer why university students' perception about GS should be studied? Why it is crucial to examine students' attitude towards UGS?

Abstract : No quantitative results presented. Needs Rewriting.

Literature Review: this section to be reduced significantly. Pl. understand it is a research paper not a report. Authors need not write too much into details of benefits of UGS/ aspects which are well established.

Figure 3.2: Is there a need to Mention QGIS software name and its version?

Methods: Too much repetition in methods. should be concise and well explained.

Section 3.2 Why Selective sampling?

Page 9 Line 276: total samples are 438 or additional samples?

Did you use simple random sampling or stratified?

Page 10 Table 1: What is the difference between item no 2 and 3 and 4 and 5?

Table 2, 3 and 4 can be merged together.

Similarly Table 8, 9, 10 and 11 can be merged together.

Authors may consider presenting some of the data through charts and graphs.

Table 12: Why one value is so extremely high?

Conclusion is weak. It needs to be revised in the light of results presented.

Discussion consist of Literature Review. it should be mixed with Introduction.

6. PLOS authors have the option to publish the peer review history of their article (what does this mean?). If published, this will include your full peer review and any attached files.

Reviewer #1: No

Reviewer #2: No

---

## [Author Response · Author response to Decision Letter 0]

22 Aug 2024

I would like to express my gratitude to the two reviewers, whose insightful remarks greatly enhanced the paper. We made every effort to respond to every remark. We really hope that the manuscript's reviewers will find it acceptable to publish.

Reviewer #1: This paper offers an important and timely contribution on the perceptions of urban green space from a developing nation's perspective. I commend the author's choice in surveying university students as younger generations are increasingly more cognizant of environmental issues and cues, and, as the author noted, are not always involved in decision-making processes.

Response: Thank you so much for your appreciation. The issue is very important and the study can be a good reference for future studies.

-While the paper has a great deal of potential and provides a much needed perspective from a developing country so acutely impacted by climate change and rapid urbanization, the paper falls short in a number of areas. First, the literature review does not provide appropriate context or definitions of typologies of green space and how this is defined in a Bangladeshi context (in Section 2.1). There was also scope to strengthen the justification for surveying university students. Please see attached document for additional recommended literature.

Response: Thank you for referring to the literature and it was very helpful in answering your comment. We have clearly defined the typologies. We also provided justification of surveying university students. Relevant information from the suggested articles have been included.

-The methodology section can be improved by justifying the quantitative approach. When investigating 'perceptions', why weren't qualitative methods such as interviews or focus groups employed to provide a more nuanced perspective. Additionally, the author states that the survey design is based on the work by Bonaiuto et al. (2003). However, there is no indication of what these perceptions are nor is this work mentioned in the Literature Review.

Response: The study conducted by Bonaiuto et al. (2003) examines the level of satisfaction that residents have with their urban districts. The unidimensional scale measuring green space exhibits strong internal consistency. Consequently, it facilitated the use of instruments that more reliable, and consistently accurate.

-Notwithstanding that the author may be from a NESB, there are several instances where meaning is lost due to grammatical errors and issues in written expression (refer to the attached document). Several assertions also require clarification. For example, in how does one know whether it’s not the wealth that leads to happiness? (Line 102) and why do university students have a greater understanding of UGS? 

Response: It implies that people living in more expensive homes would have better access to green spaces and express higher satisfaction levels.

The students of 2021-22 (1st year), 2020-21(2nd year), 2019-20 (3rd year), 2018-19 (4th year) and 2021–22 (Master’s degree program) were selected for the study because they offer diverse perception on urban green spaces. As students’ progress through their academic years, their understanding of the topic and their perceptions evolve. By including students at different stages of learning, we can gain a broader view of how perceptions of urban green spaces change over time.

-Returning to context, the paper could be further strengthened with the inclusion of a short discussion on the context of green space planning in Bangladesh. For instance, how are UGS governed and are communities consulted in decision-making processes? This would be particularly beneficial to form a more robust conclusion and set of recommendations.

Response: Updated and revised the conclusion and recommendation section

This study strongly argued the necessity of adequate planting space in metropolitan areas of Bangladesh. The management of urban green spaces in Bangladesh requires the collaboration of local and national authorities, with planning monitored by master plans and zoning regulations. Therefore, non-governmental organisations (NGOs) and community organisations at the local level must take the initiative in promoting the establishment of green spaces and pushing for the representation of community concerns in planning procedures. Furthermore, we must engage in the technical process of tree planting. Old structures pose a risk for planting trees on their rooftops. Accidents could occur if the roof gets damp. However, in the case of newly constructed buildings, cost-effective chemicals are being utilised to safeguard the roof. In addition, rather than planting trees in a random or disorganised manner, researchers and policymakers must collaborate. Collaboration with researchers generates new ideas, leading to better results.

-In summary, this paper has the potential to offer critical insights into perceptions of UGS from the developing South. However, it falls short in several areas, and thus requires major revision. I recommend that the author enhance the literature review, methodology, and English language quality and provide stronger justification for the adoption of a quantitative methodology as opposed to qualitative or mixed-methods design.

Response: Checked English and corrected to improve readability

-Reviewer #2: Authors have analyzed perception of university students for UGS however manuscript needs significant rewriting. Flow of writing is broken to that extent that not even para , each sentence is disjointed with previous sentence that makes reading not enjoyable.

Response: Checked English and corrected to improve readability

-Besides, Introduction and Literature review is written in the form of report rather than a research paper. In fact, Introduction fails to establish the need of the study. I suggest Authors to mix writeup from discussion section in Introduction to clearly bring out the need of the study. The Authors need to answer why university students' perception about UGS should be studied? Why it is crucial to examine students' attitude towards UGS?

Response: The questions are clearly answered in the introduction section and yellow marked.

Abstract: No quantitative results presented. Needs Rewriting.

Response: Rewritten

Literature Review: this section to be reduced significantly. Pl. understand it is a research paper not a report. Authors need not write too much into details of benefits of UGS/ aspects which are well established.

Response: Some sentences have been deleted from the literature review and tried to provide information concisely and coherently.

Figure 3.2: Is there a need to Mention QGIS software name and its version?

Response: The mentioned QGIS software name and its version have been deleted. One line has been added (273-274). [Note: The map is partially taken from https://www.kalerkantho.com/english/online/national/2020/04/11/34962]

Methods: Too much repetition in methods. should be concise and well explained.

Response: The repeated sentences have been deleted. Some sentences have been modified.

Section 3.2 Why Selective sampling?

Response: The confusion has been cleared. 

Page 9 Line 276: total samples are 438 or additional samples?

Response: The confusion has been cleared.

Line 298-299-Despite the sample size of 366, we increased the dataset by collecting data from an additional 72 samples. So, the total sample size is 438.

Did you use simple random sampling or stratified?

Response: The confusion has been cleared. We used random sampling. 

Page 10 Table 1: What is the difference between item no 2 and 3 and 4 and 5?

Response: The confusion has been cleared.

Item no Item

Item two: There are green spaces for relaxing

Item three: There are enough green spaces.

Item four: Green areas are in good condition

Item five: Many green areas are disappearing

Table 2, 3 and 4 can be merged together.

Response: Table 2, 3, 4 have been merged (see Table 2)

Similarly Table 8, 9, 10 and 11 can be merged together.

Response: It is difficult to merge the tables and it will be very clumsy. 

Authors may consider presenting some of the data through charts and graphs.

Response: There are so many variables in those tables. I tried in excel but couldn’t merge those charts and graphs.

Table 12: Why one value is so extremely high?

Response: The result is from analysis.

Conclusion is weak. It needs to be revised in the light of results presented.

Response: Revised and improved.

Discussion consist of Literature Review. it should be mixed with Introduction.

Response: Revised and improved, and merged with introduction.

PloS One Review – Perception of urban green space among university students in Bangladesh

Abstract – intangible services?

Response: The confusion has been cleared. 

Line 14-17. Urban management initiatives in underdeveloped nations sometimes neglect crucial services for university students, such as studying environments and recreational opportunities.

Line 57 – 59 → need connecting sentence

Response: The confusion has been cleared and showed connection. 

Line 58 – more recent data than 2016? 

Response: provided

With more than 40.47% of Bangladesh’s population living in cities as of 2023 (World Bank data), the role of urban forests is becoming increasingly critical.

63 – doesn’t make sense – how does 

Response: provided

Bangladesh's urban management strategies are significantly shaped by the impacts of climate change, particularly in relation to natural disasters such as flooding, cyclones, and rising sea levels (Dewan et al., 2021; Sultana and Selim, 2021).

78 – scope to include lit on green space perception

Response: provided

81 – really? I beg to differ!

Response: The confusion has been cleared. 

84 – enlightening uni students? Focus/justification of study unclear

Response: The confusion has been cleared. 

These difficulties may include navigating the impact of academic and social activities on the environment and finding ways to incorporate sustainable practices into their routines. They have different perspectives and experiences about green spaces and have new ideas about developing green areas in the city. By integrating perspectives from different disciplines, universities contribute to a comprehensive understanding of green space perception.

93 – a better sub-heading would be ‘Determinants of urban green space use’

Response: The confusion has been cleared. 

98 – who was surveyed? 

Response: revised and cleared. 

99 – sentence fragment

Response: corrected. 

Age, gender, level of education, and income were the only sociodemographic factors that showed differences in four parks. The users of urban green space were people over the age of 45.

102 – how does one know whether it’s not the wealth that leads to happiness?

Response: addressed and corrected. 

It implies that people living in more expensive homes would have better access to green spaces and express higher satisfaction levels.

108 – urban green what?

Response: addressed and corrected. 

114 – top concern from what?

Response: addressed and corrected. 

Parks and other public green spaces were ranked as the top concern for students regarding their preferred environments.

116 – ‘activities’ is unclear. 

Response: addressed and corrected. 

Additionally, the survey found that students used parks and green areas for activities such as eating lunch, relaxing, and studying (Speake et al., 2013).

114-130 – this seems confused – better to write thematically. 

Response: addressed and corrected. 

Parks and other public green spaces were ranked as the top concern for students regarding their preferred environments. Additionally, the survey found that students used parks and green areas for activities such as eating lunch, relaxing, and studying (Speake et al., 2013). Urban green spaces are associated with higher levels of well-being and greater aesthetic value for the elderly than for the younger generation. Adults focused on activities and cited a need for additional recreation-oriented parks for teens where students noticed maintenance issues and worried about their safety (Gearin and Kahle 2006). Those over the age of 50 had the most optimistic views, while those between the ages of 15 and 24 had the most pessimistic views. Ph.D. and professional-level respondents were the most optimistic about urban green spaces’ positive effects on air quality, while middle-school and lower-educational level respondents were the most pessimistic (Ode Sang et al., 2016). On the other hand, undergraduates are more aware than graduates (Gearin and Kahle 2006). There is a gender gap in student knowledge of urban green spaces. Because men are more likely to use parks for sports than women (Gearin and Kahle 2006). They also found that females engage in more physical activity in urban green spaces than males. In addition, females place a higher value on green spaces' aesthetic qualities and report a greater sense of well-being when they are present. Since Muslims pray five times a day, they should demand prayer rooms. Due to prayer hours, they can’t stay as long. They cannot pray by traveling from home to urban green settings. Even when mosques are closed, parkgoers choose a reserved location within the parks. They can practice their religion in urban green spaces (Duan et al., 2018) and spend time with their families.

167 – unclear

Response: addressed and corrected. 

Participants from the multi-ethnic group (British, Pakistani, Bangladeshi, Indian, Eastern European) found a link between biodiversity and their psychological healing experiences.

182 – financial returns unclear

Response: addressed and corrected. 

Green space and landscaping increase property values and land development’s financial returns. Depending on the nature of the project, research indicates that financial returns have increased anywhere from 5 to 15 percent (McMahon, 1996).

202 – developing nations, then big leap to students. A clearer connection needs to be made. 

Response: addressed and corrected. 

In recent years, scientists, citizens, and politicians have begun to explore how urban green space is perceived, especially in developing nations.

209 – space needed

Response: addressed and corrected. 

210-211 – climate change and agricultural dependence is unclear

Response: addressed and corrected. 

Bangladesh is highly vulnerable to the effects of natural disasters and climate change, which significantly affect its agriculture. As a result, the country is heavily dependent on agriculture for its economy and livelihood.

217 – research design approach (explanatory) needs to be supported by literature

Response: addressed and corrected. 

An explanatory research design is used in this study. Explanatory research is a way to find out why something happens when there isn’t much information to go on. It can help people learn more about a certain topic, figure out how or why something is happening, or even predict what will happen in the future (George, 2021). A quantitative methodology is utilized by the researcher throughout the development of this study. According to Muijs (2004), quantitative research is the process of providing explanations for phenomena through the collection and analysis of numerical data using methods that are mathematically based.

229 – conflict of interest?

Response: addressed and corrected. 

This familiarity enhances the research process by facilitating smoother communication and collaboration. Additionally, SUST was chosen based on specific criteria, such as its relevant research facilities, academic expertise, and alignment with the project’s objectives, which were evaluated to ensure the best fit for the study."

265 – why and how? A lot of unqualified assertions are being made

Response: addressed and corrected. 

The students of 2021-22 (1st year), 2020-21(2nd year), 2019-20 (3rd year), 2018-19 (4th year) and 2021–22 (Master’s degree program) were selected for the study because they offer diverse perception on urban green spaces. As students progress through their academic years, their understanding of the topic and their perceptions evolve. By including students at different stages of learning, we can gain a broader view of how perceptions of ur

---

## [Editor Report · Decision Letter 1]

12 Sep 2024

Perception of urban green space among university students in Bangladesh

PONE-D-24-18205R1

Dear Dr. Shah Md Atiqul Haq, 

We’re pleased to inform you that your manuscript has been judged scientifically suitable for publication and will be formally accepted for publication once it meets all outstanding technical requirements.

Kind regards,

Gouranga Lal Dasvarma, PhD

Academic Editor

PLOS ONE

Additional Editor Comments (optional):

Thank you for revising the manuscript by addressing the comments of the two reviewers.

I am recommending that the revised manuscript be considered for publication subject to its meeting all other conditions of PLOS ONE.
---

## [Editor Report · Acceptance letter]

20 Sep 2024

PONE-D-24-18205R1 

PLOS ONE

Dear Dr. Atiqul Haq, 

I'm pleased to inform you that your manuscript has been deemed suitable for publication in PLOS ONE. Congratulations! Your manuscript is now being handed over to our production team.

Kind regards, 

on behalf of

Dr. Gouranga Lal Dasvarma 

Academic Editor

PLOS ONE